# Promising Colorectal Cancer Biomarkers for Precision Prevention and Therapy

**DOI:** 10.3390/cancers11121932

**Published:** 2019-12-04

**Authors:** Mimmo Turano, Paolo Delrio, Daniela Rega, Francesca Cammarota, Alessia Polverino, Francesca Duraturo, Paola Izzo, Marina De Rosa

**Affiliations:** 1Department of Biology, University of Naples Federico II, 80126 Naples, Italy; 2Colorectal Surgical Oncology, Abdominal Oncology Department, Istituto Nazionale per lo studio e la cura dei tumori, ‘Fondazione Giovanni Pascale’-Istituto di Ricovero e Cura a Carattere Scientifico (IRCCS), 80131 Naples, Italy; delrio.paolo@gmail.com (P.D.); daniela.rega@gmail.com (D.R.); 3Department of Molecular Medicine and Medical Biotechnology and CEINGE Biotecnologie Avanzate, University of Naples Federico II, 80131 Naples, Italy; francesca.cammarota88@gmail.com (F.C.); ale.polverino17@gmail.com (A.P.); francesca.duraturo@unina.it (F.D.); paola.izzo@unina.it (P.I.)

**Keywords:** colorectal cancer, molecular biomarkers, cancer prevention, early cancer detection, precision therapy

## Abstract

Colorectal cancer (CRC) has been ranked as the third most prevalent cancer worldwide. Indeed, it represents 10.2% of all cancer cases. It is also the second most common cause of cancer mortality, and accounted for about 9.2% of all cancer deaths in 2018. Early detection together with a correct diagnosis and staging remains the most effective clinical strategy in terms of disease recovery. Thanks to advances in diagnostic techniques, and improvements of surgical adjuvant and palliative therapies, the mortality rate of CRC has decreased by more than 20% in the last decade. Cancer biomarkers for the early detection of CRC, its management, treatment and follow-up have contributed to the decrease in CRC mortality. Herein, we provide an overview of molecular biomarkers from tumor tissues and liquid biopsies that are approved for use in the CRC clinical setting for early detection, follow-up, and precision therapy, and of biomarkers that have not yet been officially validated and are, nowadays, under investigation.

## 1. Introduction

Colorectal cancer (CRC) is the second most common cause of cancer death, with 881,000 new deaths in 2018, and is the third most prevalent cancer worldwide, with about 1.8 million new cases in 2018. According to the GLOBOCAN 2018 database, the incidence rate of colon cancer is high in parts of Europe, while it tends to be low in most regions of Africa and Southern Asia. Notably, the incidence of CRC varies greatly between countries depending on their economic development. Arnold et al. [1] defined CRC a “socioeconomic development marker”. In fact, they divided countries into three groups based on CRC incidence and mortality: group 1, constituted by countries with a high CRC incidence and mortality, namely populations in Eastern Europe, Latin America, and Asia; group 2 constituted by countries with a high CRC incidence and a low mortality, namely European countries, Canada, and Singapore; and group 3, constituted by countries with a low CRC incidence and mortality, namely Australia, Iceland, New Zealand, and Japan, all of which have a high Human Development Index [2]. Overall, despite the increase in the incidence of CRC over the last 20 years, the incidence of CRC mortality has decreased in many countries probably due to prevention strategies, early detection and improvements in treatment [1].

Colorectal carcinogenesis is characterized by genetic and epigenetic alterations that transform normal cells into cancer cells. A characteristic of CRC is high inter- and intra-tumor heterogeneity at both clinical and molecular level. Intra-tumor variability refers to changes in distinct regions of a tumor, while temporal heterogeneity refers to changes observed overtime between the primary tumor and its matched metastases. Chromosomal instability, microsatellite instability, aberrant DNA methylation and DNA repair defects are all mechanisms that generate tumor genetic variability during colorectal epithelial cell transformation that, in turns, are responsible for patient prognosis and response to specific therapy. In the era of biological personalized care, precise molecular characterization of the tumor is crucial in defining the therapeutic plan. Consequently, the identification and standardization of cancer prognostic and predictive molecular biomarkers is becoming increasingly more relevant [3,4]. Herein we provide an overview on the approved and promising molecular biomarkers currently available for CRC. We also try to shed light on the molecular basis of CRC onset and progression, its epidemiology and principal approved therapeutic regimens in the attempt to understand the role of molecular biomarkers in the management of CRC.

In this review, we summarize bibliographic sources to analyze, interpret and critically evaluate the data available. We searched the literature related to our topic using PubMed and the PubMed Central database of the MEDLINE database, and the U.S. National Library of Medicine^®^ (NLM, Rockville Pike, Bethesda, Maryland) database. The key-words used were: “Colorectal cancer (and CRC) onset”, “Colorectal cancer (and CRC) progression”, “Colorectal Cancer (and CRC) Molecular Biomarkers”, “Colorectal cancer (and CRC) diagnostic Biomarkers”, ”Colorectal cancer (and CRC) prognostic biomarkers”, “Colorectal cancer (and CRC) predictive biomarkers”, and “Colorectal cancer (and CRC) biomarkers and therapy”. We first scrutinized the most relevant papers by the abstract, journal ranking and years of publication. We also checked the reference lists of the selected papers to identify relevant publications not found using our key-words, while also taking care to avoid duplicate citations.

## 2. Colorectal Cancer: An Overview

About 75% of CRC cases are sporadic, while only about 10% are hereditary; the remaining 10–20% are familial cases, defined as a familial cluster of CRC patients in which the genetic mechanism of onset remains unclear [5]. Usually, this group of patients, in which the disease is probably associated with low-penetrance DNA variants, does not show Mendelian inheritance, but rather high-phenotypic heterogeneity. Notably, CRC onset and progression are characterized by the well-known adenoma-carcinoma sequence.

Figure 1 shows a schematic representation of CRC onset and progression. It has been hypothesized that only cancer stem cells are able to trigger neoplastic transformation and promote tumor progression [6,7]. Epithelial cells of colorectal mucosa are organized along the crypt villus axis. At the base of the crypt are the colon stem cells, which are the more undifferentiated cells that have self-renewal and pluripotency capacity. An oncogenic hit in these cells generates a cancer stem cell, which can give rise to a cancer. An oncogenic hit in a trans-amplifying differentiated cell does not produce a cancer. However, several factors, including microenvironment factors, could induce cell de-differentiation thereby generating stem-like cells from trans-amplifying cells. An oncogenic hit in these stem-like cells could also give rise to neoplastic transformation [8].

The epithelial-to mesenchymal-transition (EMT) is a physiological process typical of epithelial cells by which the latter lose their epithelial features and acquire mesenchymal characteristics, namely motility, resistance to programmed cell death, self-renewal capability and all the features of stem cells. Mesenchymal cancer cells can alter the basement membrane components and the extracellular matrix and so trigger a metastatic process. These findings suggested that the EMT is a mechanism that generates a pool of stem-like cells that enable cancer progression, and represents a common biological mechanism that could be a target for therapeutic intervention. Ultimately, it is the combined effect of the EMT and the mesenchymal–epithelial transition (MET), that enables the metastatic progression of CRC: the EMT enables primary tumor escape and spread by way of mesenchymal intermediates, and the MET restores the CRC highly-proliferative epithelial stem cell phenotype [8]. It would be interesting to evaluate whether genes involved in the EMT and cell plasticity could serve as molecular markers in CRC follow-up and for the early detection of metastatic disease.

The main mechanisms involved in the accumulation of tumor alterations underlying cancer progression are chromosomal instability (CIN), microsatellite instability (MSI), aberrant DNA methylation (i.e., the CpG island methylator phenotype [CIMP]), and DNA repair defects [9,10,11,12]. The CIN phenotype, which is characterized by chromosome alterations, represents about 60% of all CRCs [10]. MSI, namely, the variations in the numbers of repetitive units in each microsatellite sequence, results from inactivation of the mismatch repair (MMR) system caused by mutations hitting one of the DNA *MMR* genes (i.e., *MLH1, MSH2, MSH6, PMS1, PMS2*) [13,14,15,16]. MSI accounts for only about 15–22% of sporadic CRCs, while it is a characteristic of the tumor in patients with Lynch syndrome. MSI phenotypes are distinguished based on the degree of instability: MSI-high (MSI-H) and MSI-low (MSI-L); or microsatellite stable (MSS) [9,17,18]. When the percentage of altered mono- and di-nucleotide microsatellite markers exceeds 20%, the instability is defined “High” (MSI-High). MSI is usually associated to alterations in MMR function mainly in the MSH2 or MLH1 proteins. Microsatellite alterations below 20%, which are usually found only in dinucleotide markers are referred to as “MSI-Low” [11,19,20]. Alterations of tri- and tetra-nucleotides have been associated with MSH3 dysfunction [19,20]. This kind of alteration is called “elevated microsatellite alterations at selected tetra-nucleotide repeats” (EMAST) [19,21,22,23,24]. Genes that are more often altered consequent to *MMR* inactivation, are the *TGF-β* tumor suppressor gene [25], the TGF-β type II receptor (*TGF-βR2*), *BAX*, caspase 5 apoptotic regulator [26,27], and the tumor suppressor gene *TCF4* (which has been implicated in deregulation of the Wnt/β-catenin/TCF signaling pathway) [28]. Sporadic and hereditary CRCs have different mechanisms of MMR inactivation, that mostly consist in point mutations of the *hMLH1* or *hMSH2* genes in the case of hereditary CRC, and in promoter hyper-methylation of the *hMLH1* gene in sporadic CRC [29]. Sporadic MSI CRCs are a consequent of epigenetic silencing induced by the *BRAF* V600E mutation [30]. Therefore, the latter is a diagnostic marker with which to distinguish sporadic from hereditary MSI CRC [31]. Finally, a tumor is defined “CIMP” if it shows methylation of at least three of the following markers: *CACNA1G, IGF2, NEUROG1, RUNX3 and SOCS1* [32].

The microenvironment, including the immune system and the extracellular matrix, also affects tumor heterogeneity and determines different behavior of apparently similar tumors [33,34]. Dendritic cells, tumor-associated macrophages (TAMs) and tumor infiltrating lymphocytes (TILs) are the main immunological cells involved in the host immune response to cancer cells and they have recently been identified as prognostic markers and potential targets for adjuvant therapy [35,36]. Dendritic cells are antigen-presenting cells that generate the adaptive immune response. TAMs produce components of the immunosuppressive tumor microenvironment, namely cytokines, chemokines, growth factors, and trigger T-cell activity by releasing inhibitory immune checkpoint proteins. They affect tumor progression and response to chemo-radiotherapy by acting on tumor microenvironment features [37]. Finally, TILs kill tumor cells and have thus been associated with disease outcomes [38].

Various factors contribute to the incidence of CRC. Sporadic CRCs arise consequent to somatic mutations while germline-inactivating mutations in oncogenes or tumor suppressor genes cause hereditary CRC. First-degree relatives of CRC patients have a threefold greater risk of developing CRC than individuals without familial predisposition. Patients with inflammatory bowel diseases are also at an increased risk of CRC [39]. The prognosis of CRC depends largely on the cancer stage at the time of diagnosis. The five-year survival of patients with stage I CRC is about 90% versus 10% in patients at stage IV [40].

Surgery plays a pivotal role in the treatment of patients diagnosed at an early stage of cancer. However, many patients are diagnosed at an advanced stage of disease, and sometimes have distant metastases. Adjuvant therapy may be effective in such cases, although drug resistance may affect response and concur to recurrent disease [41]. The chemotherapy approved for CRC is a combination of 5-fluorouracil and leucovorin (e.g., oxaliplatin–FOLFOX, irinotecan–FOLFIRI). In addition, two monoclonal antibodies against the epidermal growth factor receptor (cetuximab and panitumumab) are used in combination with well-established treatment regimens [42,43,44]. Biological chemotherapy also includes the vascular endothelial growth factor (VEGF)-A-targeted antibodies bevacizumab and aflibercept, which are recombinant proteins that target VEGF-A, VEGF-B and placental growth factor (PlGF) [3]. Immunotherapy results in a good response in several types of solid tumors, including CRCs. The monoclonal antibodies pembrolizumab and nivolumab, which block programmed cell death 1 (PD1), are approved by the USA Food and Drug Administration for the treatment of mismatch-repair-deficient (dMMR) and microsatellite instability-high (dMMR–MSI-H) mCRC. However, mismatch-repair-proficient (pMMR) and microsatellite instability-low (pMMR–MSI-L) CRC do not response to the immune checkpoint target therapy.

## 3. Role of Molecular Biomarkers in CRC Management

Biomarkers are defined as a multitude of biological features, such as imaging or radiomic alterations, and biological molecules found in blood and in other body fluids and tissues that are a sign of a normal or disease condition. DNA, RNA, microRNA, antibodies, and epigenetic changes are examples of biomarkers. Biomarkers play an important role in the management of CRC, indeed, they can reveal predisposition for the disease and detect the disease at an early stage. They are also useful for monitoring the efficacy of treatment, neo-adjuvant therapy, follow-up, and disease recurrence. They can also help to select the most appropriate chemotherapeutic drug across a broad spectrum of patients [41].

As we discussed previously [3], the tumor-node-metastases staging, tumor budding, and immunoscore are the best means with which to classify colon cancer and they are a guide in CRC follow-up and in therapy decision-making. With the advent of immunotherapy, CRCs are also classified based on mismatch-repair-deficiency or proficiency and the level of microsatellite instability, (dMMR–MSI-H; pMMR–MSI-L). It is now known that dMMR–MSI-H CRC is associated with a high tumor mutation burden and immune cell infiltration [45,46].

Although surgery is the gold standard treatment for early CRC, in which it can be curative, most CRC patients are diagnosed at an advanced stage [47]. The five-year survival rate after surgery of early stage (I/II) CRC patients exceeds 90% [48]. However, stage III and stage IV CRCs are characterized by local lymph node invasion and distance metastases and a very low overall survival, respectively [49,50]. This is probably because early stages of the disease are often asymptomatic and most people refuse colonoscopy and the fecal occult blood test [51]. In this scenario, it is important to identify new diagnostic and prognostic molecular biomarkers to detect the disease at an early stage to predict therapeutic response.

## 4. Molecular Features of Hereditary Colorectal Cancer

Hereditary CRC syndromes are rare diseases usually caused by germline mutations in oncogenes or in tumor suppressor genes that are crucial in such processes and events, namely colorectal mucosa turnover, cell division, cell cycle, and programmed cell death. The incidence of hereditary CRC syndromes now account for about 10% of all CRCs [52]. Notably, the prevalence of germline mutations is highest (about 16–33%) in CRC patients diagnosed before the age of 50 [53,54,55]. People with hereditary CRC syndromes show higher risk of CRC than unaffected people. They also show symptoms in other organs that are typical of each specific syndrome, often including an increased risk of extra intestinal cancer during their lifetime. The molecular diagnosis, which currently consists in the identification of pathogenic genetic variants in genes associated with both a high- and low-penetrance cancer risk, is an essential tool for cancer prevention, follow-up, counseling and survival and represents the gold standard approach in the management of these syndromes.

The hereditary syndromes predisposing to CRC are listed in Table 1. Specific genetic variants are associated with each syndrome, each with its typical onset age and responsiveness to drugs. The advent of genetic predisposition markers open the way to the prevention of cancer onset in at risk subjects, and to early cancer detection as well as to precision therapy. Interestingly, effects of NSAIDS and aspirin have long been studied to treat familial adenomatous polyposis (FAP) patients. A recent placebo-controlled randomized trial showed that treatment with a combination of sulindac and erlotinib resulted in a significant decrease of colorectal polyp onset in FAP patients after six months of treatment versus placebo [56]. The molecular identification of specific pathogenic *APC* gene variants in FAP patients revealed people with inherited disease in at-risk families, who must undergo follow-up. On the other hand, the absence of a disease-causing variant reduces the risk of CRC to that of the general population, and endoscopic surveillance could become less burdensome. Endoscopic surveillance is advisable in FAP patients without a pathogenic mutation in the *APC* gene or in one of the other genes responsible for CRC. Children carriers of an *APC* pathogenic variant should also undergo ultrasonography and alpha-fetoprotein screening protocols each 5–10 years, beginning at birth, because of the risk of hepatoblastoma is approximately 800-fold that of the general population [57].

## 5. Role of Molecular Biomarkers in the Surgical Approach to Hereditary Colorectal Cancers

Surgical options for patients with hereditary non polyposis colorectal cancer (HNPCC) range from segmental colectomy to total abdominal colectomy with ileorectal anastomosis to restorative proctocolectomy as well as to all the possible procedures for rectal cancer [68]. Identification of the specific pathogenic variants in the *MMR* genes that confirm the clinical diagnosis of Lynch syndrome could help to guide surgical decision-making. Total abdominal colectomy is considered because of the elevated risk of metachronous lesions. Patients, especially postmenopausal women, should be offered the option of prophylactically extended surgery (hysterectomy and oophorectomy) [69]. Patients with polyposis syndromes are usually offered prophylactic colectomy to prevent cancer [70]. A crucial issue is rectal sparing procedures in patients with limited rectal polyposis. Polypectomy or limited resection can be considered for patients with amartomatous polyposis syndromes. In general, surgical decision-making is based on risk factors, age of the patient, and acceptation of an intensive follow-up policy.

To our knowledge, there is no consensus as to whether genetics and molecular biomarkers could improve surgical options offered to patients with hereditary forms of CRC. The relationship between APC mutations, genotype and the severity of polyposis along the colon and in the rectum of FAP patients, has led to the hypothesis of a schematic surgical strategy, especially in terms of rectum saving procedures. Nieuwenhuis et al. showed that, despite no difference in cancer risk, the risk of deferred proctectomy after ileorectal anastomosis is increasingly higher in patients with severe polyposis [71]. However, according to Dodaro et al. [72], the decision regarding type, extension and timing of surgery should take into consideration the patient’s genotype together with her/his clinical data. The use of minimally invasive surgery has dramatically improved perioperative and long-term results [73].

## 6. Predictive Biomarkers in CRC Therapy and Prognosis

The first target of 5-FU is the thymidylate synthase (TS) protein, which is encoded by the *TYMS* gene. As expected, the response to 5-FU depends on the expression of the TS protein and of the *TYMS* gene that therefore have significant prognostic value in overall survival prediction after chemotherapy [74,75]. Moreover, the expression of molecules involved in the metabolism of 5-FU, namely thymidine phosphorylase (TP), uridine phosphorylase (UP), orotate phosphoribosyl transferase and dihydropyrimidine dehydrogenase (DPD), have been associated with the response to drugs [41]. Capecitabine is an oral drug that is converted to 5-FU consequent to the activity of the TP enzyme. Therefore, TP has prognostic value in predicting the response to capecitabine. Patients with high TP expression have a better response than patients with low TP expression, and loss of TP function causes capecitabine-resistance [76,77]. Similarly, metabolic intermediates involved in the uptake and metabolism of irinotecan, such as carboxylesterases, uridine diphosphate glucuronosyltransferase, the hepatic cytochrome P-450 enzymes CYP3A, β-glucuronidase, and the ATP-binding cassette transporter protein, are prognostic markers of response to this drug. Resistance to oxaliplatin is correlated to the expression of the nucleotide excision repair pathway [78,79].

As discussed above, the EMT and stemness confer resistance to programmed cell death to CRC cells, thereby giving rise to tumors resistant to chemoradiotherapy (Figure 1). Accordingly, stemness surface markers, such as CD133, EphB2high, EpCAMhigh, and CD44+ have been suggested as markers of colon cancer aggressiveness and resistance to therapy [80,81]. The anti-EGFR antibodies, cetuximab and panitumumab, inhibit the EGF signaling pathways thereby regulating cell proliferation. The CRYSTAL trial demonstrated the efficacy of cetuximab, in combination with a FOLFOX or FOLFIRI regimen, only in patients with CRC negative for *KRAS* or *NRAS* mutations [44,80,82] The *RAS* mutation is also a negative predictive marker for panitimumab biological therapy [83], except in the case of the G13D *KRAS* mutation, which has been associated with a positive response to the anti-EGFR antibody, comparable to that of patients with a *KRAS* wild-type tumor [84,85]. However, the prospective ICECREAM (Irinotecan Cetuximab Evaluation and Cetuximab Response Evaluation Among Patients with a G13D Mutation) study demonstrated that in patients with *KRAS* G13D-mutated chemotherapy-refractory mCRC, neithercetuximab monotherapy nor cetuximab plus irinotecan resulted in a statistically significant improvement in terms of two-year overall survival [86]. Similarly, in a meta-analysis, Rowland et al. [87] did not find any significant difference between *KRAS* G13D and other *KRAS* mutated tumors in mCRC patients treated with anti-EGFR mAbs biological therapy. Several studies have investigated the role of mutations in other genes of the EGFR pathway, namely, *PI3K, BRAF* and the quantitative expression of the PTEN protein. However, due to insufficient and/or discordant findings, those mutations are not recommended as predictive therapeutic biomarkers in clinical practice [69]. Monoclonal antibodies against vascular endothelial growth factor (VEGF) are also approved for mCRC therapy; however, their survival benefit is limited to a few months due to acquired resistance [88]. Although there are no validated predictive biomarkers relating to the use of anti-angiogenic drugs, VEGF itself has prognostic value. Indeed, high VEGF expression is associated to a poor prognosis for CRC patients, a low response to preoperative radiotherapy, and relapses. Furthermore, VEGF-C could be a prognostic biomarker in rectal cancer [89].

With regard to immunotherapy that targets the immune checkpoint, dMMR–MSI-H status is the only CRC that responds to this treatment. In this context, MMR and MSI are important predictive biomarkers for therapeutic decision-making in case of CRC, and have entered into clinical practice [90]. Interestingly, TAM infiltration is associated with a better prognosis in CRC than in other solid tumors, in which, on the contrary, TAMs have been associated with a poor prognosis [91,92,93]. Furthermore, Malesci et al. [94] observed that TAMs are positive prognostic factors for 5-FU response in stage III CRC patients. Indeed, TAM infiltration has a clear beneficial effect in patients treated with 5-FU, which has not observed in untreated patients [95,96,97,98,99,100]. Tumor infiltrating lymphocytes, and specifically the density of memory T cells (CD45RO+) in tumors, has been associated with improved survival [101]. Furthermore, in accordance with previous data showing a relationship between MSI and TILs, a high-frequency of MSI correlated with higher CD45RO+ cell density [102,103,104]. It has been suggested that MSI causes immunogenicity of tumor cells by improving the synthesis of truncated peptides [102] thereby stimulating the adaptive immune responses of mCRC. In a phase II clinical trial, the observation that the number of TILs was higher in MSI tumors than in microsatellite stable (MSS) tumors is in accordance with this hypothesis [99,105,106]. As expected, MSI-H is an important predictive biomarker with which to select patients who may benefit from immunotherapy. Indeed, treatment with pembrolizumab (anti-PD-1) and nivolumab (anti-PD-L1) resulted in a better objective response, stable disease, and progression-free survival in MSI-H patients, but not in MSS mCRC patients. In addition, the levels of PD-1 and PD-L1 were significantly higher in dMMR tumors than in proficient MMR (pMMR) tumors. These observations led to the approval of immunotherapy for these MSI patients [107]. Furthermore, measurement of the tumor mutation burden in the primary tumor and/or in blood samples from melanoma or lung cancer patients has been suggested as a biomarker of therapeutic efficacy of the immune checkpoint inhibitors [108,109]. The presence of a high number of tumor-associated neoantigens could improve the identification of cancer cells by the immune system. In this respect, MSI-H CRCs are correlated with increased infiltration of TILs, such as the CD8+ cytotoxic lymphocytes, which are Th1-activated cells that produces IFNγ, and CD45 RO+ T memory cells, which, in turn, are also correlated with a better survival versus MSS CRC [110,111,112,113].

## 7. Future Perspectives in the Field of Cancer Biomarkers

### 7.1. Molecular Subtypes

Next-generation sequencing spurred a broad spectrum of data regarding the molecular characterization of solid tumors, including CRC. The CRC Subtyping Consortium classified CRC into subgroups based on a common molecular “core signature” [114]. They identified four consensus molecular subtypes (CMS) and defined the biological features of each subtype. The features of CMS1 are hypermutated phenotype, MSI and CIMP phenotype with *BRAF* mutations, immune infiltration, and shorter post-relapse survival. CMS1 has been defined “MSI-immune” and accounts for about 14% of CRCs. Conversely, CMS2, CMS3, and CMS4 show high CIN phenotype. CMS2, which is the canonical subtype, represents about 37% of all CRC cases and is characterized by high somatic copy number alterations (SCNAs) and by activation of the WNT and p53 pathways. CMS3, the metabolic subtype, accounts for about of 13% of all CRCs and is characterized by metabolic deregulations, *KRAS* mutations, a mixed MSI status, SCNA and CIMP low. Finally, CMS4, which is the mesenchymal subtype, represents about 13% of all CRCs and is characterized by TGF-beta activation, angiogenesis, stromal infiltration, high SCNA, and worse relapse-free and overall survival [114].

Notably, the CMS classification has been proposed as a predictive factor for chemotherapy response in mCRC. Indeed, in a retrospective study, both progression-free and overall survival were better in CMS4 patients treated with an irinotecan regimen in first-line therapy than in those treated with oxaliplatin chemotherapy. On the other hand, in CRC patients undergoing EGFR treatment, the worse progression-free and overall survival occurred in CMS1 and the best in CMS2 patients [115]. Similarly, in an in vitro study, 5-FU induced high apoptosis in cancer cell lines belonging to CMS subtypes 1 to 3, and low or no apoptosis in CMS4 cells [116]. Furthermore, the response to oxaliplatin was poor or absent in CMS4 cells, and the best in CMS2 cells [117].

Another CRC classification is based on the CRC intrinsic subtype (CRIS) that consists in the features own of the patient’s colon cancer cells, not affected by their non-neoplastic tissue components, primarily cancer-associated fibroblasts (CAFs), that are a strong indicator of tumor aggressiveness [118,119].

The authors defined its role as a prognostic and predictive biomarker. In accordance with this classification, the CRIS-A subtype is constituted by *BRAF*-mutated-MSI and *KRAS*-mutated-MSS tumors. Although these tumors are unresponsiveness to the therapy now available, it is conceivable that they could respond to anti-metabolic therapies that are now under investigation because they have strong glycolytic/hypoxic features [120]. CRIS-B tumors are characterized by activation of TGF-beta signaling and EMT program and by high invasiveness and a poor prognosis. However, they are unrelated to the CMS4 mesenchymal subtype, which has the same features, but is of stromal origin. The CRIS-C subtype is constituted by tumors with elevated EGFR signaling and sensitivity to EGFR inhibitors, independently of all known gene mutations. The CRIS-D subtype is constituted by tumors that activate the WNT pathway and in which IGF2 is overexpressed, which probably induces resistance to biological therapy with EGFR antibody [121]. Finally, CRIS-E is constituted by tumors with high WNT pathway activation, a Paneth cell-like phenotype and *KRAS* mutations, and are thus resistant to anti-EGFR antibody treatment. The CRIS components -C, -D, and -E are characterized by high WNT pathway activity, which suggests they could benefit from drugs targeting this pathway [122,123].

However, the CRIS tumor categorization classifies the tumor taking into account only the specific features of cancer cells, whereas the relevance of the stromal compartment is well-known in cancer aggressiveness, progression, and response to therapy. Thus, the integration of stromal signatures, mainly CAF infiltration and CRIS traits results in a tumor classification more powerful in terms of prognosis and prediction than CRIS traits or CAFs alone. For example, patients with low CAF infiltration and non-CRIS-B subtype have a good prognosis and do not require adjuvant chemotherapy, while, patients with low CAF infiltration and CRIS-B subtype have a poor prognosis and are predicted to be unresponsive to traditional chemotherapy. However, it is conceivable that patients in this group could benefit from drugs that target the TGF-β pathway and that are now under investigation [124].

In our opinion, the classification of CRC molecular subtypes can easily be applied in clinical practice. Moreover, confirmation and validation of these concepts and findings will lead to a more precise understanding of the role and power of each molecular subtype as a prognostic and predictive tool for the management of CRC.

### 7.2. Circulating Biomarkers

The term “liquid biopsy”, initially referred to the detection of circulating tumor cells (CTCs) [125], whereas it now refers to the detection of many tumor traits in the peripheral blood of patients [126,127,128]. Circulating tumor DNA (ctDNA), CTCs, exosomes, and microRNAs present in the bloodstream of patients are considered promising biomarkers for the management of CRC. Circulating cell-free DNA (cfDNA) present in blood and other body fluids are produced from cellular apoptosis, necrosis, phagocytosis, and active secretion [129]. ctDNA is the fraction of cfDNA that originates from tumor cells; it can easily be quantified by digital PCR on small volumes of plasma, and can rapidly identify somatic tumor mutations [130,131,132]. It has been suggested that the increased levels of ctDNA in the blood of advanced and metastatic cancer patients versus the ctDNA level observed in early stage cancer patients [133], may account for the tumor burden [134,135]. The presence of ctDNA in the peripheral blood of patients can be used to determine genotypic changes that occur during systemic treatment, and that can render such therapy ineffective [136]. In this context, serial ctDNA measurements can reveal the response of mCRC patients to treatment, which suggests that ctDNA could be an early predictor of treatment response, to complement the standard Response Evaluation Criteria In Solid Tumors-based disease assessment [137], and a guide for anti-EGFR therapy [138]. Moreover, the presence of ctDNA in the peripheral blood of CRC patients, negatively impacts on their survival and it has been considered a prognostic factor in clinical studies [139,140].

CTCs are cells that, after undergoing the EMT, have detached from the primary tumor and are shed daily into bloodstream at a rate of approximately 10 million cells per tumor gram [141,142]. However, due to platelet cloaks or coagulation factors that surround CTCs, a fraction of cells elude detection and are found in a low concentration in peripheral blood [143]. Although many CTC detection methods have been described, only the Cell Search System (Veridex LLC, Raritan, NJ) has been approved by the US Food and Drug Administration for CRC and for breast and prostate cancer [144]. As described for ctDNA, peripheral blood CTCs were reported to be of considerable importance in early stage and metastatic cancer. CTC evaluation is a non-invasive procedure with which to diagnose cancer at an early stage [145] and a useful prognostic factor for cancer progression and survival [126]. Notably, CTCs proved to be a prognostic marker in cases of mCRC, in which levels of CEA and other markers were not measurable [141]. Moreover, elevated CTC levels were associated with worse clinical outcome parameters, overall survival and progression-free survival in CRC patients [141,146,147].

Exosomes are small cellular vesicles, spontaneously released by many cell types. They contain protein and nucleic acid and are involved in both physiological and pathological processes. Exosomes derived from CRCs have been implicated in such tumor processes as EMT [148], migration [149], and metastasis [150]. In recent years, many attempts have been made to identify diagnostic, prognostic, and treatment response biomarkers in CRC exosomes. Some studies focused on isolating miRNAs from tumor exosomes as potential biomarkers for the detection of CRC disease [151]. Other studies have been conducted on serum miRNAs, which however seem to be less stable than the exosomal miRNA [152]. Ogata-Kawata and colleagues [153] showed that serum levels of seven miRNAs (let-7a, miR-1229, miR-1246, miR150, miR-21, miR-223, and miR-23a) were significantly higher in CRC patients than in healthy controls, which indicates that these miRNAs may detect CRCs. In addition, the sensitivity of miR23a and miR-1246 was much higher than that of the CA19-9 and CEA markers for stage I CRC, which again suggests that these miRNAs are potential biomarkers for the detection of early stage CRC [153].

## 8. Conclusions

Colorectal cancer is a heterogeneous disease, characterized by inter- and intra-tumor variability. Molecular alterations accumulate in the colorectal mucosa via various mechanisms, i.e., MMR gene alterations, chromosomal instability, and CpG island methylation alterations, which, in turn, lead to cancer onset and progression. All these mechanisms confer specific features to each tumor in terms of malignancy, aggressiveness, invasion and response to therapy. Notwithstanding the increase in CRC, its mortality has decreased probably due to prevention approaches, early detection and improvements in therapeutic strategies. Precision therapy, based on the tumor’s molecular features, is often combined with chemoradiotherapy. In this context, molecular biomarkers, defined as biological molecules that are a sign of a normal or tumor condition, and also of tumor predisposition, play a crucial role.

An overview of diagnostic, prognostic, and predictive biomarkers that could be used in the management of CRC is provided in Table 2.

Several molecular biomarkers have been approved for use in clinical practice and are essential tools that support therapeutic decisions. This is the case of *KRAS* mutations, *BRAF* mutations and MSI/MSS status. On the other hand, germline genetic variants in specific disease-causing genes are associated to hereditary CRC syndrome, which strongly suggests tumor predisposition. High throughput screening technology has produced a large quantity of data. The classification of these data is shedding light on the nature of tumors, and could, in the near future, upturn the clinical and therapeutic approaches to CRC. This applies also to the classification of molecular subtypes. A better classification and validation of molecular subtypes could help to improve the outcome of precision therapy by providing information about the cancer that single molecular biomarkers alone could not provide.

Finally, molecular circulating biomarkers are promising tools in the management of CRC. Liquid biopsies can be used in cancer screening, and to determine the tumor burden and residual disease. They are also prognostic and predictive biomarkers, not all of which have been approved for clinical practice. Consequently, given the large body of evidence of the efficacy of these biomarkers, a concerted effort should be made to validate them for the benefit of patients.

## Figures and Tables

**Figure 1 cancers-11-01932-f001:**
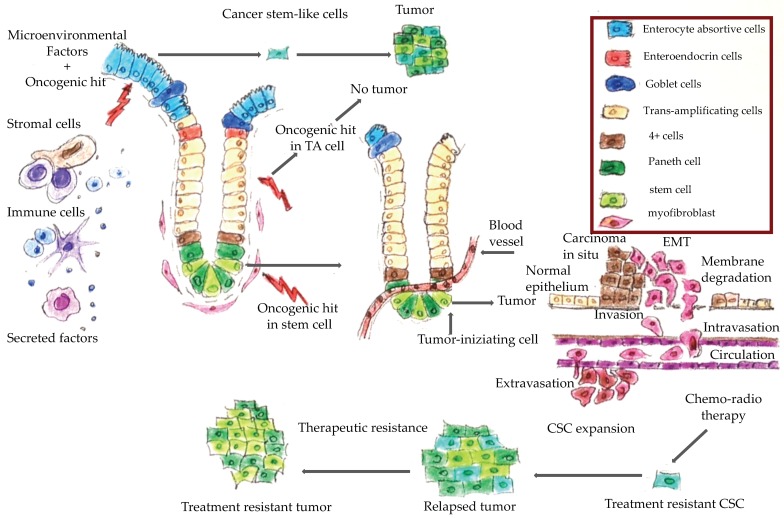
Colorectal cancer tumorigenesis. During tumor progression, epithelial cancer cells undergo the epithelial-to mesenchymal-transition (EMT) program that is characterized by acquisition of mesenchymal and stem-like cell properties consequent to which the cancer cells can invade the extracellular matrix and migrate into the surrounding tissues. They then join the endothelial cells from vessels and arrive in the lumen in a process known as ‘intravasation’. These cells can survive in the vessel lumen, then exit the vases (i.e., ‘extravasation’), disseminate into the adjacent organs, and colonize them to generate micrometastases. Chemo-radiotherapy often kills differentiated cancer cells, while the mesenchymal, stem-like cells are treatment-resistant and can give rise to a treatment-resistant tumor.

**Table 1 cancers-11-01932-t001:** Features of hereditary colorectal cancers.

Syndrome	Genes	Inheritance	Recomended Age of Screening	Tumor Molecular Features
**Adenomatous Polyposis Syndromes**				
FAP/AFAP	*APC*	Autosomal dominant	20/10–12 years	CIN, *APC* mutations [58].
PPAP	*POLE, POLD1*	Autosomal dominant	none	Controversial percentage of G > T/C > A transversions [59].
MAP	*MUTYH*	Autosomal recessive	30–50 years [60]	*KRAS, p53, APC* mutations [57].
NAP	*NTHL1* [61]	Autosomal recessive	none	none relevant
MSH3 polyposis	*MSH3*	Autosomal recessive	none	EMAST, MSI-L [62].
**Amartomatous Polyposis Syndromes**				
PJS	*STK11*	Autosomal dominant	10–15 years [60]	none relevant
PHTS	*PTEN*	Autosomal dominant	none	none relevant
JPS	*BMPR1A, SMAD4*	Autosomal dominant	15 years or earlier [63]	none relevant
**Mixed Polyposis**				
HMPS	*GREM1, BRAF*	Autosomal dominant	none	*BRAF* and *KRAS* mutations, MSI [64].
**Serrated Adenomas**				
SPS	*RNF43*	Autosomal dominant	none	*BRAF* V600E and *KRAS* (codons 12 and 13) mutations, MLH1 methylation, MGMT methylation, CIMP [65].
**Nonpolyposis CRC**				
LYNCH	*MSH2, MLH1, MSH6, MSH3, PMS2, EPCAM*	Autosomal dominant	20–25 (ten years earlier than the youngest age of colon cancer diagnosis in the family)	MSI-H, MSI-L, EMAST V600E *BRAF* wt [66].
NONPOLYPOSIS CRC-MSS	*RPS20*	Autosomal dominant	none	MSI-*BRAF* mutations *LINE-1* methylation, V600E *BRAF* wt [67].

FAP: familial adenomatous polyposis; PPAP: polymerase proofreading-associated polyposis; MAP: MUTYH associated polyposis; NAP: NTHL1-associated polyposis; PJS: Peuts–Jeghers syndrome; PHTS: PTEN hamartoma tumor syndrome; JPS: juvenile polyposis syndrome; HMPS: hereditary mixed polyposis syndrome; SPS: serrated polyposis syndrome.

**Table 2 cancers-11-01932-t002:** Diagnostic, prognostic and predictive biomarkers in CRC management.

Biomarkers	Diagnostic Value	Prognostic Value	Predictive Value
**CIN phenotype**	*APC* mutated sporadic and hereditary CRC	marker of poor prognosis	Identify high-risk patients with stage II CRC who might benefit from adjuvant chemotherapy [9,154]
**CIMP**	Specific of serrated adenomas	Marker of poor prognosis	conflicting data exsist [155]
**MSI**	Lynch syndrome [14] Sporadic MSI tumor in combination with *BRAF* V600E mutation	MSI-H is associated with better prognosis and survival versus MSI-L and MSS [112]	MSI-H is associated with worse response to 5-Flurouracil-based chemotherapy compared to MSI-L and MSS [112]; dMMR–MSI-H is associated with good renponse to immunotherapy [107].
***BRAF* V600E mutation**	Sporadic MSI CRC [27]; serrated polyposis syndrome [63]	none suggested	none suggested
***KRAS* mutation**	none suggested	marker of poor prognosis.	Identify patients resistant to anti-EGFR antibody treatment [82].
**VEGF**	none suggested	marker of poor prognosis	
**TAMs**	none suggested	marker of good prognosis [69]	Identify patients who can benefit from treatment with 5-FU [91,92,93,94,95,96].
**TILs**	none suggested	marker of good prognosis and survival [100]	Identify patients who can benefit from immunotherapy [96,101,105].
**CAFs**	none suggested	marker of tumor and aggressivenes and poor prognosis in untreated CRC	none suggested
**TS protein and *TYMS* gene expression**	none suggested	High TS and *TYMS* expression correlates with good overall survival after chemotherapy	High TS and *TYMS* expression are associated with good response to 5-FU [76,77].
**TP protein**	none suggested	none suggested	High TP expression is associated with good response to capecitabine; loss of TP function causes capecitabine-resistance [76,77].
**CMS1 (MSI-Immune)**	none suggested	none suggested	Identify patients with poor progression-free and overall survival after EGFR treatment [115]; good response to 5-FU treatment is suggested by in vitro study [116].
**CMS2 (canonical subtype)**	none suggested	none suggested	Identify patients with poor progression-free and overall survival after EGFR treatment [115]; good response to 5-FU treatment is suggested by in vitro study [116]; Identify patients with the best responce to oxaliplatin [117].
**CMS3 (metabolic subtype)**	none suggested	none suggested	good response to 5-FU treatment is suggested by in vitro study [116].
**CMS4 (mesenchymal subtype)**	none suggested	none suggested	Identify patients with better progression-free and overall survival with an irinotecan regimen than with oxaliplatin chemotherapy [115]; poor or absent responce to oxaliplatin [117].
**ctDNA**	Allows identification of genotypic changes that occur during systemic treatment [136]	Marker of poor survival [139,140]	serial ctDNA measurements could be an early predictor of treatment response [139,140].
**CTCs**	Marker of both early stage and metastatic cancer [145]	Marker of worse clinical outcome parameters, overall survival and progression-free survival [141,147,148]	none suggested
**Circulating exosomal miRNAs**	Marker of early detection [153]	none suggested	none suggested

CIN: chromosomal instability; CIMP: the CpG island methylator phenotype; MSI: microsatellite instability; VEGF: vascular endothelial growth factor; TAMs: tumor-associated macrophages; TILs: tumor infiltrating lymphocytes; CAFs: cancer-associated fibroblasts; TS: thymidylate synthase; TP: thymidine phosphorylase; CMS: consensus molecular subtypes; ctDNA: Circulating tumor DNA; CTCs: circulating tumor cells.

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
