# Peer review of "Promising Colorectal Cancer Biomarkers for Precision Prevention and Therapy"

_cancers, 2019, doi:10.3390/cancers11121932_

Round 1

Reviewer 1 Report

The manuscript is highly improved. 

Reviewer 2 Report

The new version has acquired clearness and synthesis according to the indications. The manuscript is now suitable for publication.

This manuscript is a resubmission of an earlier submission. The following is a list of the peer review reports and author responses from that submission.

Round 1

Reviewer 1 Report

line 82 please provide reference.

This is a interesting and broad review; however, no information related to the methodology used are provided. please add.

Author Response

Comments to the Author

line 82 please provide reference.

Our reply: As suggested, we now cite two references for this statement in paragraph 2 (page , lines 80).

This is a interesting and broad review; however, no information related to the methodology used are provided. please add.

Our reply: We thank the Reviewer for this comment and now describe the methods we used at the end of the Introduction section (page 2, lines 61-70)

Reviewer 2 Report

The manuscript by Turano et al covers an important field of research; in fact, the identification of markers for personalized treatments in colorectal cancer gives the hope to devise a more effective and less burdensome use of therapies.

However, the manuscript in the present form aims to cover too many different aspects, somehow loosing focus and clearness. Restricting the discussion to few most recent issues may be more useful for the reader to gain an insight into the state of the art in the field. For example, introductive aspects on epidemiology, tumorigenesis and therapy, likely well known to the reader, may be shrink in a short paragraph, and Fig. 1 may be eliminated. The chapter scheme may also be reconsidered, as the discussion about markers of different nature (i.e. mutational status, membrane-expressed antigens or cellular features) is dispersed, and so is the discussion of hereditary vs sporadic cancers, or of the prognostic vs therapeutic and surgical markers. The mentioning of circulating markers on the abstract is misleading, since it turns out not to be a focus of the review. Indeed, a table summarizing the most important markers/approaches covered by the review may also add clarity.

Accordingly, some of the references are redundant and/or not quite recent or relevant. Reference paragraph in any case should be carefully checked, since for example Ref 5 seems to be misplaced, while Ref 3 and Ref 44 correspond to the same paper.

Editing by a mother tongue may also help in order for the writing flow to gain synthesis and clarity, and to settle some sentences sounding incomplete, such as in line 287-288, page 8.

Statement regarding NRAS as a therapeutic predictive marker, at lines 300-304 page 8, should be modified, since NRAS testing is now required prior of anti EGFR therapy, see for example: (https://www.ema.europa.eu/en/documents/product-information/erbitux-epar-product-information_en.pdf)

Author Response

Reviewer 2

Comments to the Author

Restricting the discussion to few most recent issues may be more useful for the reader to gain an insight into the state of the art in the field. For example, introductive aspects on epidemiology, tumorigenesis and therapy, likely well known to the reader, may be shrink in a short paragraph, and Fig. 1 may be eliminated.

Our reply: We take the reviewer’s point, and with the help of native English editor, have largely rewritten various sections of the article to make it more focused and to improve readability. As requested, we have summarized the introductive aspects into a single short paragraph. We would prefer to retain Figure 1 because it represents a schematic representation of colorectal cancer progression that is needed to understand the role of molecular biomarkers in the management of CRC.

The chapter scheme may also be reconsidered, as the discussion about markers of different nature (i.e. mutational status, membrane-expressed antigens or cellular features) is dispersed, and so is the discussion of hereditary vs sporadic cancers, or of the prognostic vs therapeutic and surgical markers. The mentioning of circulating markers on the abstract is misleading, since it turns out not to be a focus of the review.

Our reply: As mentioned above, we have largely rewritten the text also in view of these comments. In addition we have eliminated from the abstract, the sentences about circulating biomarkers.

A table summarizing the most important markers/approaches covered by the review may also add clarity.

Our reply: We now provide a summarizing table at the end of the last section (11-13)

4.Some of the references are redundant and/or not quite recent or relevant. Reference paragraph in any case should be carefully checked, since for example Ref 3 and Ref 44 correspond to the same paper.

Our reply: As suggested, we have revised the reverences for redundancy and year of publication.

Editing by a mother tongue may also help in order for the writing flow to gain synthesis and clarity, and to settle some sentences sounding incomplete, such as in line 287-288, page 8.

Our reply: As suggested, we did editing the text by a mother tongue and it is now more fluid.

Statement regarding NRAS as a therapeutic predictive marker, at lines 300-304 page 8, should be modified, since NRAS testing is now required prior of anti EGFR therapy

Our reply: As suggested, we modified the statement regarding NRAS as a therapeutic predictive marker, at page 8 lines 276. The text now is: “The CRYSTAL trial demonstrated the efficacy of cetuximab, in combination with a FOLFOX or FOLFIRI regimen, only in patients with CRC negative for KRAS or NRAS mutations [84, 86-88]”.

Reviewer 3 Report

In the review article titled “Title: Promising colorectal cancer biomarkers for  precision prevention and therapy”  De Rosa and co-authors summarize validated and not yet validated molecular biomarkers for early detection of colorectal cancer achievable from tumour tissues and liquid biopsies.

The review article addresses a really important issue and is of interest for various professional figures, i.e. researchers, clinicians, et cet.

Scientific literature mining is comprehensive and up-to-date

The figures are explicatory

MINOR CONCERN

There are many typos. I recommend accurate professional proofreading of the manuscript with proper corrections  

Author Response

Reviewer 3

Comments to the Author

There are many typos. I recommend accurate professional proofreading of the manuscript with proper corrections

Our reply: as suggested, the article has been edited by a professional native English editor, who is a longstanding member of the European Association of Science Editors.